# Viral metagenomics in mosquitoes as potential vectors of arboviruses in the Colombian Caribbean: characterisation of a "core" regional RNA virome

**Richard Hoyos-López**[1]/+, **Daniel Echeverri-De la Hoz**[1,2], **Caty Martínez-Bravo**[1], **Bertha Gastelbondo-Pastrana**[1], **Maira Alemán-Santos**[1], **Evelin Garay**[1], **Yesica López**[1], **Héctor Contreras**[1], **Ketty Galeano**[1], **German Arrieta**[1], **Salim Mattar**[1]

[1]Universidad de Córdoba, Instituto de Investigaciones Biológicas del Trópico, Córdoba, Colombia
[2]Universidad de Santander, Facultad de Ciencias Médicas y de la Salud, Grupo de Investigación en Ciencias, Valledupar, Cesar, Colombia

**BACKGROUND** Mosquitoes are critical vectors in tropical regions where arboviruses like dengue and Zika are prevalent. This study focuses on characterising the RNA virome of mosquitoes in the Colombian Caribbean, emphasising the core regional virome and its role in the dynamics of arboviruses.

**OBJECTIVES** The objective was to identify and analyse the core RNA virome of mosquitoes across different genera and seasons in the Colombian Caribbean to understand its composition and potential influence on arbovirus transmission dynamics.

**METHODS** In 2023, 4,074 mosquitoes from the genera *Mansonia*, *Coquillettidia*, and *Anopheles* were collected across Córdoba, Sucre, Bolívar, and Magdalena during rainy and dry seasons. Specimens were pooled in groups of 50, subjected to RNA extraction, and sequenced on the MGI-G50™ platform. Bioinformatic analyses utilised the DIAMOND-MEGANizer pipeline and R packages (phyloseq, vegan, ggplot2) to identify viral communities.

**FINDINGS** The analysis identified 22 viral families and 24 unclassified RNA viruses. The core regional virome, consistently present across species and seasons, was dominated by insect-specific viruses (ISVs) such as *Aedes aegypti* to virus 1 and 2, Astopletus, and Cumbaru, alongside Picornaviridae (30% of reads), Rhabdoviridae (20%), Orthomyxoviridae, and Bunyavirales. *Mansonia titillans* (38 species) and *Coquillettidia nigricans* (21 species) exhibited the highest viral richness. No significant arboviruses were detected, highlighting ISV dominance. Virome composition varied seasonally, with greater diversity in the rainy season due to increased breeding site availability and temperature.

**MAIN CONCLUSIONS** The stability of the core virome suggests it modulates vector competence, potentially reducing arbovirus transmission. These findings advocate the use of metagenomics for enhanced vector surveillance and biological control strategies in neotropical ecosystems.

Key words: viral metagenomics - next generation sequencing - mosquitoes - arboviruses - virus diversity

Colombia is a country with diverse ecosystems that support endemic zones for a wide range of mosquito species and hematophagous habits, allowing the spread of diseases, including malaria and arboviruses such as dengue (DENV), yellow fever (YFV), Venezuelan equine encephalitis (VEEV), and other viruses with historically low transmission rates.[1] Many of these arboviruses, which are transmitted by mosquitoes, pose significant public health challenges, particularly in tropical and subtropical areas.[2] In Colombia, the primary families of medically significant arboviruses include Flaviviridae (*e.g.*, *Flavivirus*), *Togaviridae* (*e.g.*, *Alphavirus*), and *Bunyaviridae* (*e.g.*, *Orthobunyavirus* and *Phlebovirus*).[3,4] Arboviruses, such as the West Nile virus (WNV), Saint Louis encephalitis virus (SLEV), Zika virus (ZIKV), and Chikungunya virus (CHIKV), are frequently associated with human diseases in South America. Suspected vectors of arboviruses in Colombia include genera such as *Aedes*, *Anopheles*, *Culex*, and *Haemagogus*, as well as members of the tribe *Mansoniini* and *Sabethini*, which have demonstrated potential roles in arbovirus transmission[3,4,5] and are associated with preserved ecosystems and rural areas.[6,7,8,9,10] However, due to ecological changes, such as habitat fragmentation,[10] the diversity of bats, rodents, primates, avians, marsupials,[11,12,13,14] and urban expansion,[13] there is an increased frequency of human-vector contact, increasing the likelihood of outbreaks of emerging and re-emerging arboviruses.[11,12,13,14]

Financial support: This research was financed by the Minister of Science, Technology and Innovation of Colombia – MINCIENCIAS (Research Project 601 de 2022 code 91722: "Fortalecimiento de las capacidades de investigación en salud pública: metagenómica de agentes infecciosos en mosquitos y quirópteros de cuatro departamentos del Caribe").
+ Corresponding author: richardhoyosl@correo.unicordoba.edu.co | ⓘ https://orcid.org/0000-0003-1195-681X

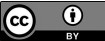

Recent advances in next-generation sequencing (NGS) and metagenomic approaches have transformed the detection of viruses within mosquito populations,[15] the discovery of new viruses,[16] the monitoring of emerging and re-emerging arboviruses,[17] and the enabling of comprehensive virome profiling of individual vectors.[18] Advances in viral metagenomics applied to mosquitoes have led to the discovery of a vast number of viruses, expanding our understanding of their diversity, classification, and the various environmental conditions in which these viral agents can persist in these vectors.[19] In addition to arboviruses, a representative group within the mosquito virome, known as insect-specific viruses (ISVs), has been identified.[20,21] These viruses naturally infect arthropods but cannot replicate in vertebrate cells or infect humans, suggesting a long-term symbiotic relationship with their hosts.[22] In fact, it has been proposed that ISVs play a significant role in modulating vector competence and could be a key component in the design of new strategies for arbovirus control.[23] Batson et al. used metagenomic sequencing to detect 24 known viral species and 46 novel species in mosquito populations, revealing a diverse viral community that included members of *Reoviridae*, *Picornavirales*, and *Flaviviridae*.[24] Other studies have applied NGS techniques to analyse viral diversity and identify numerous viral families within mosquito populations across different regions, including *Flaviviridae*, *Togaviridae*, *Phasmaviridae*, and *Phenuiviridae*.[25] These findings underscore the potential of metagenomic tools for the early detection of both known and novel arboviruses with zoonotic potential.[19,21]

Although metagenomic surveillance of arboviruses has been widely used in several countries, its application in Colombia remains relatively limited.[9,26] Although specific viruses have been genomically monitored in some Colombian mosquito populations,[14,27] comprehensive virome studies targeting potentially zoonotic viruses circulating in Colombia, particularly in the Caribbean, are sparse.[9,26] The Colombian Caribbean is a strategic region for arbovirus research due to its high diversity of mosquito vectors — including *Aedes*, *Anopheles*, *Culex*, *Mansonia*, and *Coquillettidia* — and the heterogeneity of its habitats, such as wetlands, mangroves, gallery forests, agricultural areas, and peri-urban environments.[3,6,8,10,12,14] These ecological conditions favour the persistence and transmission of emerging and re-emerging viruses, and facilitate interactions between mosquitoes and a wide range of potential reservoirs, including migratory birds — implicated in the spread of WNV and SLEV across the continent —, bats, rodents, and other wild mammals.[1,3,5,8,10-14] In this sense, the departments of Cordoba, Sucre, Bolivar, and Magdalena, combine factors that make them a priority for entomovirological surveillance: (i) a history of circulation of arboviruses other than DENV, ZIKV, and CHIKV, including serological and molecular reports of WNV, SLEV, and YFV; (ii) the presence of aquatic ecosystems and flood-prone areas that sustain large mosquito populations; (iii) intense interaction between natural areas and agricultural or livestock landscapes, increasing human-vector contact; and (iv) their location within an ecological and migratory connectivity corridor that may facilitate the introduction and spread of pathogens.[28,29]

Worldwide studies have shown that metagenomics is a valuable tool for viral surveillance and biodiversity assessment, demonstrating advantages over traditional virus detection methods in identifying a broader range of viruses.[15-19,21,24] In tropical settings such as Colombia, metagenomics holds great promise for monitoring the dynamics, diversity, and ecology of mosquito-borne viruses. Conducting virome studies using this metagenomic approach in this Colombian region could provide valuable data on viral diversity and possible emerging and re-emerging viral pathogens, which is fundamental to viral surveillance strategies and could contribute to early detection efforts and strengthen preventive public health measures to mitigate the risk of arbovirus outbreaks in the country.

This study aimed to characterise the RNA virome of mosquito vectors of arboviruses, such as *Mansonia titillans*, *Coquillettidia nigricans*, *Anopheles albimanus*, *Anopheles darlingi*, *Culex nigripalpus*, and *Culex quinquefasciatus*, which are highly abundant in ecosystems related to the Colombian Caribbean, through metagenomic sequencing.

## MATERIALS AND METHODS

*Sampling sites* - The sites included Moñitos (Córdoba) (9°14'51.2"N 76°07'31.2"W), Colosó (Sucre) (9°29'30.7"N 75°21'07.6"W), Talaigua Nuevo (Bolívar) (9°18'08.4"N 74°33'59.0"W), and Santa Ana (Magdalena) (9°19'15.3"N 74°34'33.2"W). These sites were selected based on a combination of ecological, epidemiological, and land-use criteria. Ecologically, they encompass a variety of habitats — including wetlands, mangroves, gallery forests, agricultural lands, and peri-urban areas — that sustain high mosquito diversity (*Aedes*, *Anopheles*, *Culex*, *Mansonia*, *Coquillettidia*) and provide breeding conditions favourable for arbovirus circulation. Epidemiologically, these departments have documented the presence of arboviruses beyond DENV, ZIKV, and CHIKV, including serological and molecular evidence of WNV, SLEV, and yellow fever virus (YFV), reflecting their potential as hotspots for emerging and re-emerging vector-borne pathogens.[3,5,10]

The selected sites are also situated along important migratory bird routes and contain diverse vertebrate fauna such as bats, rodents, and other mammals, which may act as reservoirs or amplifying hosts for arboviruses.[11,12,13] Anthropogenic factors, including wetland fragmentation, rice and monoculture expansion, and cattle ranching, increase human–vector contact and may enhance the likelihood of viral spillover.[3,5] Additionally, the departments' land-use patterns, agricultural activity, and presence of aquatic ecosystems prone to seasonal flooding create ideal conditions for sustaining large mosquito populations.[14]

Vector sampling was conducted between February and October 2023, with two main field campaigns in each department, timed to coincide with the transitional periods before and after the rainy and dry seasons char-

acteristic of the Caribbean climate. The dry-to-rainy season transition was sampled between February and April 2023, while the rainy-to-dry season transition was sampled between September and October 2023.

In each location, the sampling effort included eight CDC light traps operated for 12 h during the night (18:00-06:00), complemented by active searches carried out by teams of three-four trained personnel using mouth aspirators at potential resting sites among vegetation during early morning (06:00-10:00) and late afternoon (15:00-18:00) hours. Additionally, a Shannon trap was operated from 18:00 to 21:00, during which the same three-four trained personnel performed active mosquito collections using mouth aspirators to capture the specimens attracted to the light.

*Preservation of field-collected mosquito specimens* - Insect capture was performed using CDC light traps, manual capture was carried out with entomological nets and suction pumps from 07:00 to 17:00 for diurnal species, and Shannon traps were placed between 18:00 and 22:00 with light-emitting diode (LED) lights. The captured specimens were separated from non-Culicidae insects, stored in cryovials to the lowest possible taxonomic category, and transported in liquid nitrogen to the Tropical Biological Research Institute (IIBT) at the University of Córdoba, Colombia. Upon arrival, the samples were organised into cryoboxes and stored at -60°C to -80°C until taxonomic identification.

*Taxonomic identification of mosquitoes* - Identification was carried out using specialised taxonomic keys for Neotropical Culicidae[30-39] in a climate-controlled room (16°C) under stereomicroscopes, with specimens placed on chilled trays to preserve their morphological integrity. Mosquitoes were identified to the lowest possible taxonomic level, representative specimens of each morphospecies were set aside and re-examined to verify key diagnostic characters, ensuring consistency and accuracy across all samples. For morphologically challenging groups (*e.g.*, *Culex* subgenus *Melanoconion*), identification included the evaluation of multiple diagnostic traits, and when male specimens were available, genitalia were dissected and analysed to confirm species-level assignments. Although morphological identification served as the primary method for host confirmation, metagenomic sequencing data were also screened for mosquito-specific genetic markers. A subset of quality-filtered reads from each pool was aligned (data no shown) against reference mitochondrial cytochrome oxidase I (COI) sequences and whole-genome scaffolds of relevant mosquito species available in the NCBI database. This cross-validation step was particularly useful for species complexes and ecologically overlapping taxa.

Mosquitoes were pooled by species, considering factors such as sampling date, season (rainy or dry), location, and trap type. Each sample was further separated by daytime or night-time capture, habitat characteristics (riverbanks, residential and peri-domestic areas, and vegetation presence or absence), and trap type, allowing detailed data on the location of mosquito populations

and potential viral vectors. The identified females were separated into 1.5 mL tubes per municipality and species to form pools of 50 individuals, which were stored at -80°C until nucleic acid extraction.

*RNA Extraction, library preparation, and sequencing* - Mosquito pools of 50 individuals were used to ensure sufficient viral RNA detection and high-quality metagenomic libraries. Pooling enhances sensitivity for low-abundance viruses, which may be below detection limits in individual mosquitoes, and provides adequate RNA yield for robust sequencing. The mosquito cells were triturated in cold mort µL of Dulbecco's minimum essential medium supplemented with 10% foetal bovine serum (FBS) and 1% penicillin and clarified by centrifugation at 13000 rpm for 30 min. From the resulting supernatant (400 µL), was filtered through a 0.22 µm membrane, and nucleic acids were extracted using the MagMAX™ Viral/Pathogen Nucleic Acid Isolation Kit (Thermo Scientific, Waltham, Massachusetts, United States), following the manufacturer's instructions. For sequencing, the RNA concentration and integrity number (RIN) were measured fluorometrically using a QubitTM Broad Range (BR) RNA Quantification Kit and an RNA IQÔ Assay Kit (Thermo Fisher Scientific). An input of 250-500 ng total RNA was used for library preparation. Samples were processed using the MGIEasyÔ Fast RNA library preparation set and high-throughput DNA nanobead (DNB) technology. RNA was fragmented into approximately 250 nucleotides using the FCL 150 paired-end (PE) platform. The first and second DNA strands were synthesised using random hexamer primers. The fragments were subjected to end-catalytic repair (ERAT), molecular barcode ligation, and product amplification using polymerase chain reaction (PCR). The library concentration was determined using the QubitTM dsDNA Quantification Assay Kit, and the fragment size was determined using a Fragment AnalyzerTM (Agilent Technologies). Pools were obtained for DNA circularisation and DNB synthesis (> 11 ng/µL), based on the concentration and size of the fragments. Next-generation sequencing was performed using DNB on an MGI-G50TM equipment (Shenzhen, China).

*Bioinformatic analysis* - This viral metagenomic analysis was conducted using the DIAMOND-MEGANizer pipeline,[40,41] providing an efficient approach for taxonomic and functional classification of large metagenomic datasets by combining DIAMOND's rapid sequence alignment capabilities with MEGANizer's precise taxonomic binning. Initially, raw reads were preprocessed by quality filtering and trimming using Fastp to eliminate adapters and low-quality bases (minimum quality score of 20 and a read length of 50 bp). Subsequently, Bowtie2 was used to remove host genome contamination, allowing the focus to remain on the viral and microbial sequences. *De novo* assembly, with a minimum length of 300 nucleotides, was performed using the MEGAHIT software.[41] Subsequently, the filtered reads were aligned to the NCBI non-redundant (nr) protein database using DIAMOND in the BLASTx mode.

DIAMOND was configured with an E-value threshold of 1e-5, limiting target sequences to one to focus on the best hit and using a sensitive mode for optimal capture of diverse viral sequences. The alignment output was saved in DAA format (DIAMOND Alignment Archive (DAA) format, which is ideal for downstream taxonomic analysis using MEGANizer.

The MEGANizer module of MEGAN6 was used to taxonomically classify DIAMOND alignment results. MEGANizer employs NCBI taxonomy and mapping files to assign reads to taxonomic groups and applies the Lowest Common Ancestor (LCA) algorithm to ensure robust assignments. Customised filtering thresholds, such as a minimum bit score of 50 and requiring at least 50% read coverage, enhanced the accuracy of the assignment. These taxonomic classifications provided an in-depth profile of the viral and microbial community composition, identifying the prevalent viral families and genera.

Diversity and abundance metrics were calculated to analyse the viral community structures across the samples. The R programming language with packages such as phyloseq, vegan, and ggplot2 was used to visualise the taxonomic and functional profiles.[42] To ensure robust and accurate diversity estimates, raw sequence data were quality-filtered and normalised prior to analysis. Alpha diversity metrics were calculated for within-sample viral richness and evenness. Beta diversity was assessed using Bray-Curtis dissimilarity indices to measure differences in viral community composition between samples. The Bray-Curtis dissimilarity index was chosen for beta diversity to account for both presence/absence and relative abundance of viruses, critical for viromes with high abundance variation and sparsity.

Alpha and beta diversity metrics allowed for comparisons of viral diversity across samples, localities, mosquito species, and sampling periods. Hierarchical clustering and heatmaps further highlighted the similarities and differences in viral assemblages among the samples. Additionally, taxon abundance tables were generated to quantify the presence and distribution of viral families, allowing for a comparative analysis between mosquito species and environmental conditions. The final visualisations displayed taxonomic summaries, functional analyses, and abundance heatmaps, facilitating the interpretation of viral dynamics in environmental samples.[43]

## RESULTS

A total of 4,074 mosquitoes were collected, and pools were organised into 33 samples corresponding to the mosquito species previously selected based on abundance (Table). 1,729 were captured during the dry season and 2,345 during the rainy season. The most abundant species identified across the four locations in the Colombian Caribbean included *Ma. titillans*, *Cq. nigricans*, and *An. albimanus*, other relevant species were *Anopheles darlingi*, *Cx. nigripalpus*, and *Cx. quinquefasciatus*.

Bioinformatics analyses revealed contigs associated with 22 viral families and 24 RNA viruses with no current taxonomic classification in the study mosquitoes. ISVs are the most notorious and abundant viruses among the mosquito species studied. In this sense, *Ma. titillans*

presented 38 viral species, of which 17 had no taxonomic classification, while 21 viruses belonged to the orders Picornavirales, Bunyavirales, and Mononegavirales, but could not be placed within the families of these orders (Fig. 1). Viruses from the families Picornaviridae and Rhabdoviridae were also identified in the present study. *Cq. nigricans* presented 21 viral species; seven were identified as RNA viruses with no known taxonomic classification, whereas the remaining belonged to the families Orthomyxoviridae, Parvoviridae, Baculoviridae, Nudiviridae, Flaviviridae, Totiviridae, and Metaviridae and the orders Bunyavirales and Mononegavirales.

*Culex quinquefasciatus* and *Cx. nigripalpus* shared a high diversity of unclassified riboviruses and viruses; in fact, *Cx. nigripalpus* is the mosquito species with the most unclassified viruses, with only the Rhabdoviridae family being identified. *Cx. quinquefasciatus* belong to the Metaviridae and Mesoniviridae families. Viral diversity in *An. triannulatus*, *An. darlingi* and *An. albimanus* showed similar viral groups represented by the Artoviridae, Baculoviridae, Metaviridae, and Mesoniviridae families, with Rhabdoviridae viruses detected in *An. manus* (Fig. 1).

The ISV most representative was *Aedes aegypti to virus 1* and *Ae. aegypti*, and *virus 2* was identified in all mosquito species studied, followed by Astopletus, *Gordis, Cumbaru*, *Kaiowa*, *Keturi*, *Nefer*, *Nejeret*, and *Wilkie ophio-like virus 1* (Fig. 2). Other viruses with low abundance included *Culex flavivirus*, *Chibugado virus*, and *Atrato Partititi-like virus*.

Regarding viral ecology, when comparing the alpha diversity found in the localities, similarities were observed between Santa Ana/Talaigua Nuevo, Moñitos, and Colosó, along with a segregation in viral diversity results between the dry/rainy seasons, and mosquito genus (Fig. 3). This can be explained by changes in the diversity and abundance of mosquito species collected during both entomological sampling periods (Fig. 4). However, non-metric Multidimensional Scaling (NMDS) of the viral communities found by sampling period and location highlighted a 'core' of similarity among the viruses present in the different evaluated mosquito species (Fig. 5).

## DISCUSSION

In this study, we analysed the virome of mosquitoes captured in the Colombian Caribbean, where ISVs were the most abundant viruses among all mosquito species analysed. Although these viruses do not pose a direct risk to humans, they can influence mosquito vector competition and the transmission of arboviruses of public health importance. Future research should evaluate the influence of ISVs on vector capacity and their implications for developing mosquito-borne disease control strategies.

Analysis of viral diversity showed that *Ma. titillans* and *Cq. nigricans* harbored the highest viral richness compared with other species. First, the viability of using populations of mosquito species in viral metagenomics was demonstrated, which can provide much more accurate RNA virome profiles, allowing interspecific comparisons. In the present study, similarities were observed between *Ma. titillans* and *Cq. nigricans* in terms of viral

TABLE

Samples sequenced by mosquito species and summary quality data of metagenomic analysis results

| Pool (Sample) | Mosquito species | Number of raw reads | Classified reads | | Unclassified reads | reads with low quality: | | Viral contigs | mean length before filtering | mean length after filtering |
|---|---|---|---|---|---|---|---|---|---|---|
| | | | Number | % | Number | Number | % | Number | | |
| Pool 1 | *Mansonia titillans** | 63.545616 M | 63.520688 M | 99.960771% | 24.928 | 12.108000 K | 0.019054% | 25 | 150bp | 147bp |
| Pool 4 | *Ma. titillans* | 42.891598 M | 42.868378 M | 99.945864% | 23.220 | 6.258000 K | 0.014590% | 135 | 150 bp | 146 bp |
| Pool 5 | *Ma. titillans* | 39.183604 M | 39.161586 M | 99.943808% | 22.018 | 7.540000 K | 0.019243% | 191 | 150 bp | 146 bp |
| Pool 6 | *Coquillettidia nigricans* | 32.510398 M | 32.485280 M | 99.922739% | 25.118 | 12.782000 K | 0.039317% | 190 | 150 bp | 146 bp |
| Pool 7 | *Ma. titillans* | 34.086056 M | 34.068206 M | 99.947633% | 17.850 | 5.004000 K | 0.014680% | 159 | 150 bp | 145 bp |
| Pool 8 | *Ma. titillans* | 47.286180 M | 47.260626 M | 99.945959% | 25.554 | 7.874000 K | 0.016652% | 227 | 150 bp | 147 bp |
| Pool 9 | *Ma. titillans* | 39.441954 M | 39.420532 M | 99.945687% | 21.422 | 9.008000 K | 0.024248% | 370 | | |
| Pool 10 | *Ma. titillans* | 39.441954 M | 39.420532 M | 99.945687% | 21.422 | 6.578000 K | 0.016678% | 196 | | |
| Pool 11 | *Ma. titillans* | 14.408840 M | 14.356840 M | 99.639110% | 52.000 | 50.816000 K | 0.352672% | 102 | 150 bp | 149 bp |
| Pool 12 | *Ma. titillans* | 26.695796 M | 26.615892 M | 99.700687% | 79.904 | 77.848000 K | 0.291611% | 118 | 150 bp | 147 bp |
| Pool 13 | *Ma. titillans* | 31.219236 M | 31.139840 M | 99.745682% | 79.396 | 77.108000 K | 0.246989% | 118 | 150 bp | 146 bp |
| Pool 14 | *Ma. titillans* | 24.188584 M | 24.123974 M | 99.732891% | 64.610 | 62.916000 K | 0.260106% | 82 | 150 bp | 146 bp |
| Pool 15 | *Cq. nigricans* | 23.446948 M | 23.381252 M | 99.719810% | 65.696 | 64.140000 K | 0.273554% | 75 | 150 bp | 144 bp |
| Pool 16 | *Cq. nigricans* | 31.440184 M | 31.377876 M | 99.801820% | 62.308 | 59.964000 K | 0.190724% | 25 | 150 bp | 143 bp |
| Pool 17 | *Cq. nigricans* | 29.150906 M | 28.958474 M | 99.339876% | 192.432 | 191.758000 K | 0.657811% | 382 | 150 bp | 148 bp |
| Pool 18 | *Ma. titillans* | 18.289456 M | 18.100306 M | 98.965798% | 189.150 | 188.612000 K | 1.031261% | 239 | 150 bp | 146 bp |
| Pool 19 | *Ma. titillans* | 13.203390 M | 13.079460 M | 99.061377% | 123.930 | 123.588000 K | 0.936032% | 250 | 150 bp | 146 bp |
| Pool 21 | *Anopheles albimanus* | 21.596384 M | 21.477542 M | 99.449713% | 118.842 | 118.276000 K | 0.547666% | 74 | 150 bp | 148 bp |
| Pool 22 | *An. albimanus* | 17.133586 M | 17.042842 M | 99.470374% | 90.744 | 90.196000 K | 0.526428% | 53 | 150 bp | 148 bp |
| Pool 23 | *Culex nigripalpus* | 22.466642 M | 22.392558 M | 99.670249% | 74.084 | 73.462000 K | 0.326983% | 14 | 150 bp | 148 bp |
| Pool 24 | *Cx. nigripalpus* | 22.861688 M | 22.721658 M | 99.387491% | 140.030 | 139.476000 K | 0.610086% | 50 | 150 bp | 148 bp |
| Pool 25 | *Anopheles darlingi* | 35.837204 M | 35.631098 M | 99.424883% | 206.106 | 205.106000 K | 0.572327% | 86 | 150 bp | 148 bp |
| Pool 26 | *Anopheles trianulatus* | 3.523866 M | 3.507178 M | 99.526429% | 16.688 | 16.588000 K | 0.470733% | 26 | 150 bp | 147 bp |
| Pool 30 | *Ma. titillans_* | 41.013562 M | 40.948962 M | 99.842491% | 64.600 | 63.888000 K | 0.155773% | 374 | 150 bp | 148 bp |
| Pool 31 | *Culex quinquefasciatus* | 23.133230 M | 23.062480 M | 99.694163% | 70.750 | 70.130000 K | 0.303157% | 44 | 150 bp | 147 bp |
| Pool 33 | *Cx. quinquefasciatus* | 44.969288 M | 44.886730 M | 99.816412% | 82.558 | 81.908000 K | 0.182142% | 167 | 150 bp | 147 bp |
| Pool 34 | Cx. quinquefasciatus | 39.372892 M | 39.316710 M | 99.857308% | 56.182 | 55.750000 K | 0.141595% | 39 | 150 bp | 147 bp |
| Pool 35 | *Cx. quinquefasciatus* | 44.903198 M | 44.824108 M | 99.823866% | 79.090 | 78.554000 K | 0.174941% | 60 | 150 bp | 147 bp |
| Pool 36 | *Cx. nigripalpus* | 49.382146 M | 49.319874 M | 99.873898% | 62.272 | 61.622000 K | 0.124786% | 78 | 150 bp | 147 bp |
| Pool 37 | *Cx. nigripalpus* | 52.652456 M | 52.580268 M | 99.862897% | 72.188 | 71.618000 K | 0.136020% | 84 | 150 bp | 146 bp |
| Pool 38 | *Ma. titillans* | 45.698564 M | 45.659744 M | 99.915052% | 38.820 | 38.258000 K | 0.083718% | 509 | 150 bp | 148 bp |

*Species names are presented in full at their first mention. RNA: ribonucleic acid.

abundance and diversity, particularly for Bunyavirales, Metaviridae, and viruses that were not systematically classified. However, for *Cq. nigricans*, differences were observed among the families Flaviviridae, Parvoviridae, Orthomyxoviridae, Baculoviridae and Nudiviridae. These two mosquito species appear to have their own relatively stable "eukaryotic virome," despite the geographical distances in the study areas, which could have significant implications for their ability to transmit medically important arboviruses to humans.

Exploring the mosquito virome and understanding how its composition influences arbovirus transmission is essential for understanding the emergence of arboviral diseases and the dynamics of their outbreaks. In this study, we performed metagenomic sequencing to characterise the virome of epidemiologically important mosquito species in the Colombian Caribbean region, which has varying ecological conditions that may affect vector-virus interactions and viral diversity. The mosquito species analysed were selected based on their abundance across different localities and seasons, including *Ma. titillans*, *Cq. nigricans*, *An. albimanus*, *An. darlingi*, *Cx. nigripalpus*, and *Cx. quinquefasciatus*.

Our analysis revealed a notable predominance of ISVs in the viromes of all mosquito species studied. These ISVs, which infect insects but not vertebrates,

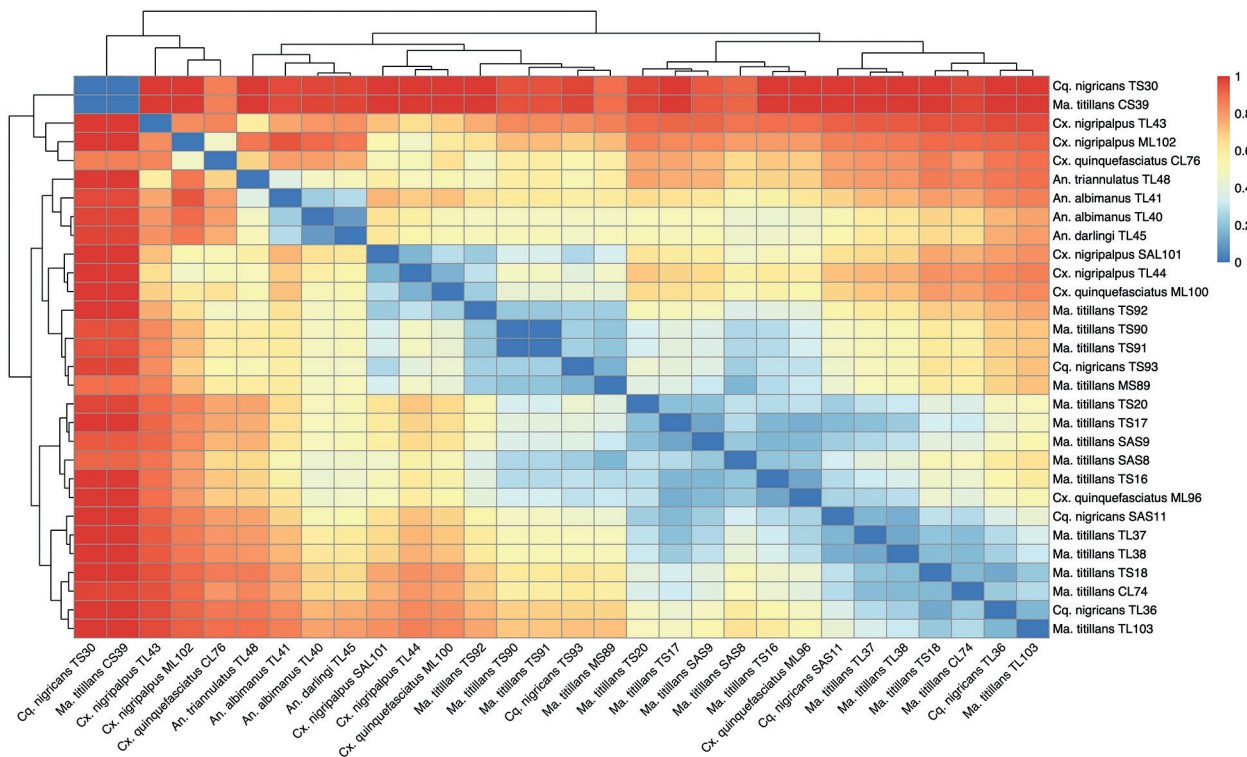

Fig. 1: relative abundance of viral families in mosquito species. Bar graph showing viral families in the virome of each mosquito species: *Mansonia titillans*, *Culex quinquefasciatus*, *Culex nigripalpus*, *Coquillettidia nigricans*, *Anopheles triannulatus*, *Anopheles darlingi*, and *Anopheles albimanus*. ISVs: insect-specific viruses. The abundance of viral families was estimated by transforming the number of reads into relative values, which provided an assessment of their presence in each mosquito species.

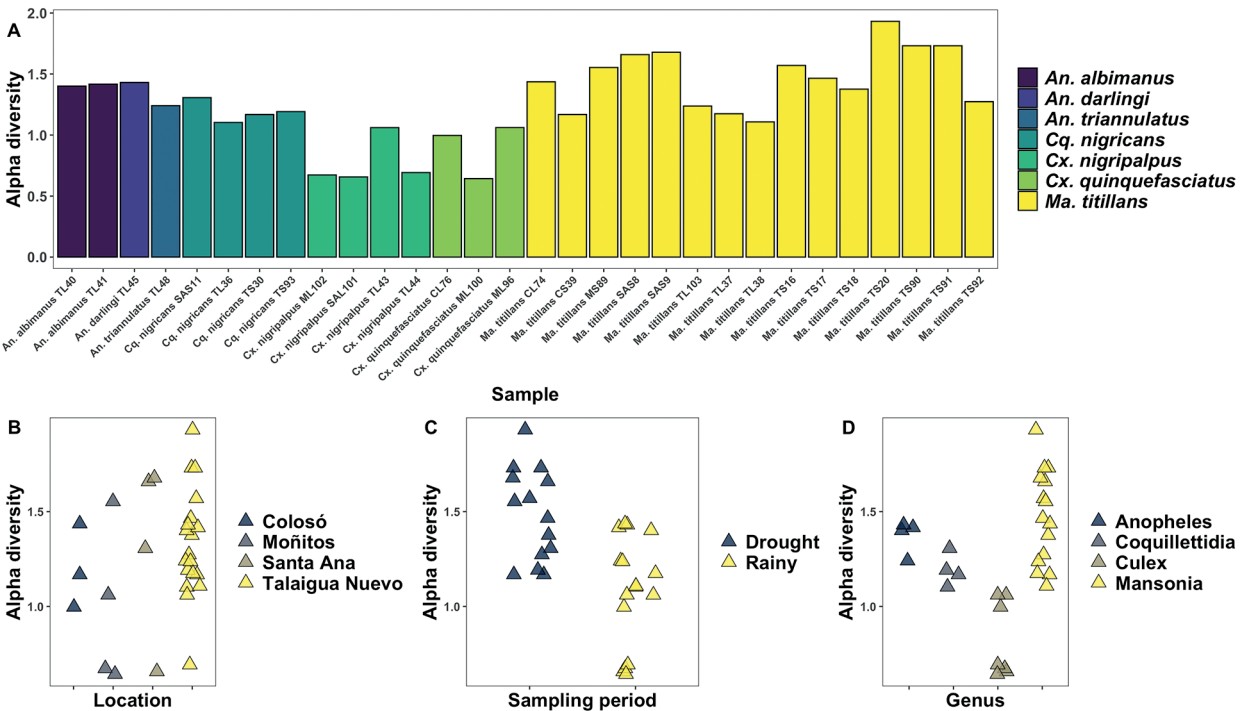

Fig. 2: relative abundance of insect-specific virus (ISVs) in mosquito species. Bubble plot showing the relative abundance of ISVs. The abundance of the viral species was estimated by transforming the number of reads into relative values, providing an assessment of their presence in each mosquito species.

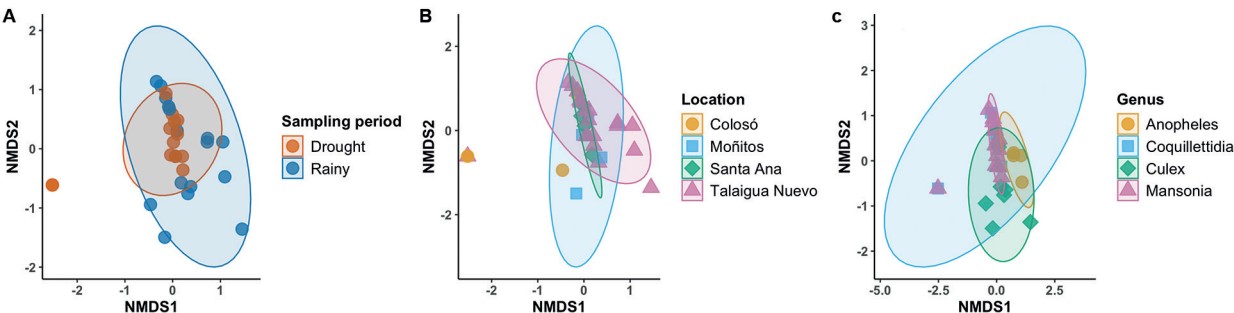

Fig. 3: estimation of Alpha diversity using the Shannon-Wiener index (H') and non-metric multidimensional scaling (NMDS) for the viral diversities found according to mosquito species, sampling locality, and sampling period.

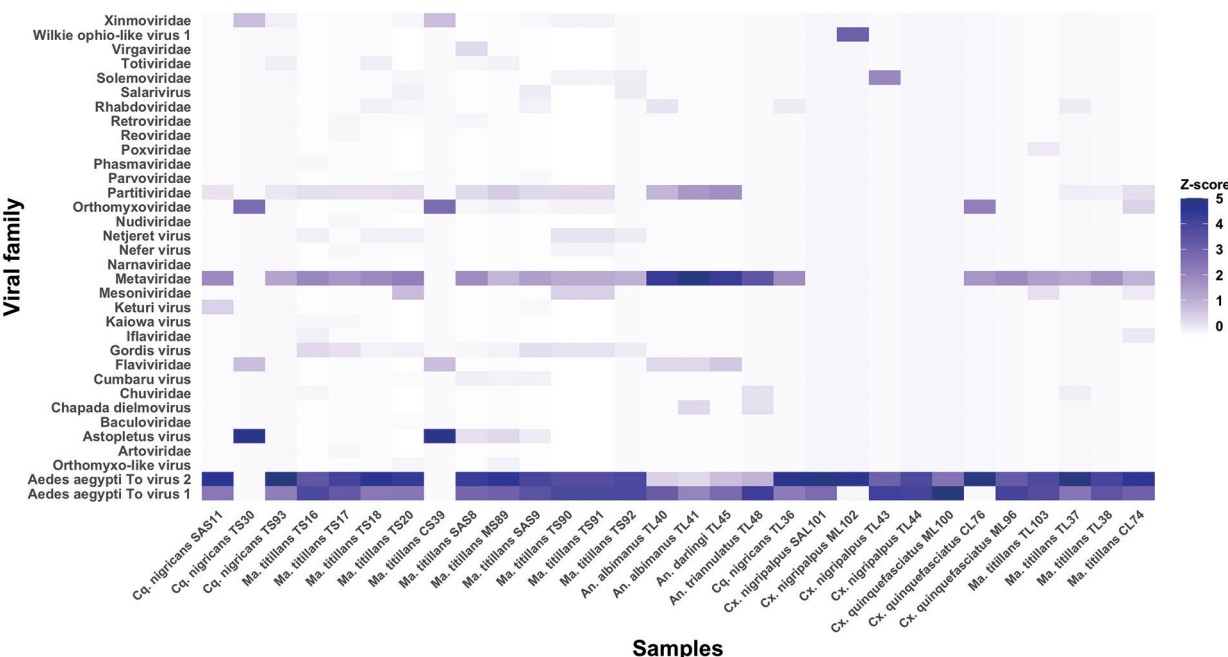

Fig. 4: non-metric multidimensional scaling (NMDS) between sampling period, location, and mosquito genus.

have been identified as potential modulators of arboviral transmission. The presence of ISVs, such as *Ae. aegypti* to virus 1 and *Ae. aegypti* to virus 2 in all mosquito species studied, underscores the importance of these viruses in shaping the dynamics of the mosquito virome. Additionally, ISVs such as *Astopletus*, *Gordis*, *Cumbaru*, *Kaiowa*, *Keturi*, *Nefer*, *Nejeret*, and *Wilkie ophio-like virus 1* been detected with varying prevalence across species. Interestingly, *Cx. flavivirus*, *Chibugado virus*, and *Atrato Partititi-like viruses* were found in lower abundance, suggesting potential geographical or ecological constraints on their distribution. Taxonomic analysis of viral sequences identified a high diversity of viral families associated with mosquitoes, including Picornaviridae, Rhabdoviridae, Orthomyxoviridae, Parvoviridae, Baculoviridae, Nudiviridae, Flaviviridae, Totiviridae, and Metaviridae, as well as unclassified RNA viruses. Notably, *Ma. titillans* harbored the greatest viral diversity, with 38 viral species, including 17 with no taxonomic

classification. This highlights the need for further investigation to elucidate the evolutionary relationships and ecological roles of these viruses. Similarly, *Cq. nigricans* exhibits diverse viromes, including members of the orders Bunyavirales and Mononegavirales, which contain arboviruses of medical importance. The high abundance of unclassified Riboviria viruses in *Cx. nigripalpus* suggests a potential reservoir function for novel viruses that requires further investigation.

In previous virome and metagenomic studies on mosquitoes, various viral families were identified in different mosquito species. The specific findings for each species studied in our research and their comparisons with studies in other regions are presented below.

*Mansonia titillans* - In this study, 38 viral species were identified, including members of the families Picornaviridae, Rhabdoviridae, Orthomyxoviridae, Parvoviridae, Baculoviridae, Nudiviridae, Flaviviridae, Totiviridae, and Metaviridae. Previous studies in Brazil

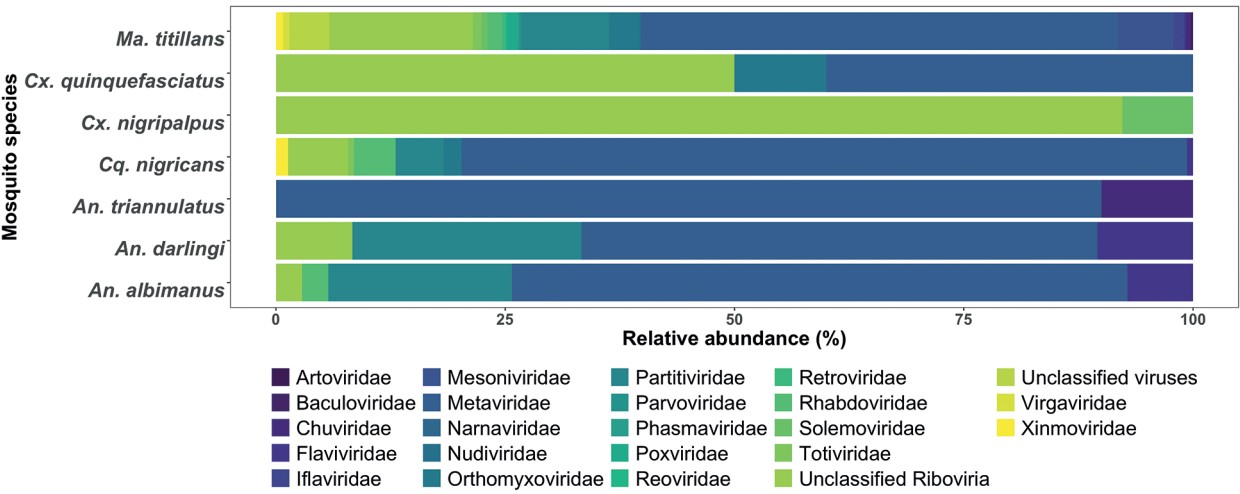

Fig. 5: heatmap matrix using a colour scale; the similarity between the viral diversities found in the sequenced samples of each mosquito species.

have reported the presence of Flaviviridae and Bunyavirales in this species, as well as viruses such as the *Mansonia* flavivirus.[44,45,46,47]

*Coquillettidia nigricans* - Viruses from the orders Bunyavirales and Mononegavirales were detected. In studies from other regions, such as Argentina, Flaviviridae and Rhabdoviridae viruses have been identified, with reports of Coquillettidia-associated viruses.[48,49,50,51]

*Anopheles albimanus* - In this study, viruses from the families Flaviviridae and Rhabdoviridae were identified. Previous studies have documented the presence of Mesoniviridae and Peribunyaviridae in this species in Mexico and Colombia, including Anopheles-associated flaviviruses.[45,47,52-59]

*Anopheles darling* - Primarily flaviviruses and a small number of unclassified viruses were detected. Studies in Brazil and Peru have identified viruses from Totiviridae and Iflaviridae, respectively.[44,50]

*Culex nigripalpus* - A high abundance of unclassified Riboviria was observed in this species. In studies from North America and Brazil, Rhabdoviridae, Flaviviridae, and Peribunyaviridae have been reported, including *Cx. nigripalpus* nucleopolyhedrovirus.[48,54,58]

*Culex quinquefasciatus* - Viruses from Flaviviridae and Rhabdoviridae were detected in this study. Studies from Asia and Africa have recorded Totiviridae, Iflaviridae, and Baculoviridae in this species, with the presence of Culex-associated flaviviruses and Culex-borne rhabdoviruses.[46,53,56]

The observed differences in viral diversity across mosquito species and localities may be driven by environmental factors such as temperature, humidity, and breeding site availability [Supplementary data (Table)]. Our findings indicate similarities in alpha diversity between the mosquito viromes from Santa Ana/Talaigua Nuevo, Moñitos, and Coloso, whereas a clear segregation was observed between the viromes of mosquitoes collected during the dry and rainy seasons. Alpha diversity analysis indicated significant variability between localities and seasons, highlighting the influence of environmental factors on the structure of the virome. In addition, the segregation observed in the beta diversity analysis (NMDS) suggests that viral composition varies between species and seasons, with an increase in viral richness in the rainy season. This seasonal effect aligns with previous studies, indicating that fluctuations in mosquito population dynamics influence the virome composition. Additionally, NMDS analysis revealed a 'core' of similarity among viral communities across different mosquito species, supporting the idea of shared viral reservoirs and horizontal transmission of certain ISVs in the studied region. Similarly, the results showed that Santa Ana and Talaigua Nuevo share a homogeneous viral profile, possibly due to ecological similarities and the presence of nearby bodies of water. Colosó presents a differentiated viral community, which could be due to habitat and the availability of alternative hosts.

From an epidemiological perspective, the presence of diverse viral families, particularly *Rhabdoviridae*, *Flaviviridae*, and *Mesoniviridae*, highlights the potential of mosquitoes to serve as vectors for emerging viruses. Given that *Culex* and *Anopheles* mosquitoes are known to transmit arboviruses of medical importance, their high viral diversity warrants continued surveillance to detect potential spillovers. Moreover, the detection of ISVs with a broad geographic distribution suggests that these viruses may play a role in vector competence, possibly influencing mosquito susceptibility to pathogenic arboviruses.

The implications of these findings extend to vector control strategies and biotechnology. The potential of ISVs as biological control agents has gained attention in recent years because they may interfere with the replication of pathogenic viruses within mosquitoes through mechanisms such as competitive exclusion and immune priming in the host. The identification of common ISVs across multiple vector species reinforces the need to explore their ecological roles and interactions with arboviruses in the future. Additionally, the discovery of unclas-

sified viruses in this study highlights the importance of continued metagenomic research to expand our knowledge of mosquito-associated viromes and their potential applications in vector-borne disease management.

*In conclusion* - Our study provides an in-depth characterisation of the mosquito RNA virome in the Colombian Caribbean region, revealing a rich diversity of ISVs and unclassified RNA viruses. The seasonal and geographic variations observed in viral diversity emphasise the need for longitudinal studies to assess virome stability over time. Further investigations into the ecological and evolutionary dynamics of mosquito-associated viruses are crucial for understanding their roles in arbovirus transmission and vector ecology. Our findings provide valuable insights into the virome composition of Neotropical mosquito species and highlight the potential of ISVs as tools for future vector-control strategies. Future research should focus on the functional characterisation of unclassified viruses, particularly those detected in mosquito species that are known viral pathogen vectors, such as species of the genus *Culex*. In addition, the integration of metagenomic data with ecological and epidemiological studies will allow a broader understanding of the transmission dynamics of arboviruses in the region, development of preventive strategies, and strengthening of the capacity to respond to outbreaks of mosquito-borne diseases in the Colombian Caribbean.

## ACKNOWLEDGEMENTS

To the public health authorities of the municipalities and their communities in the study location.

## AUTHORS' CONTRIBUTION

SM and JDR designed the initial study; RHL, DEH, CMB, MAS, EG and KG carried out the fieldwork; RHL and MAS identified mosquitoes and prepared pools for RNA extraction; RHL, DEH, CMB, MAS, EG, KG, YL and HC performed RNA extraction, library preparation, and RNA sequencing; DEH, MAS and RHL implemented the bioinformatic analyses; SM and RHL wrote the first draft of the manuscript. All authors contributed to data interpretation, critically revised the manuscript, and approved the final version.

## DATA AVAILABILITY

The sequences are available in the Sequence Read Archive (SRA) under the accession number PRJNA1251058.

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

# OPEN PEER REVIEW

Memórias do IOC thanks the anonymous reviewers for their contribution to the peer review of this work.

## FIRST REVIEW ROUND

REVIEWERS' COMMENTS

### REVIEWER #1

Reviewer comments:

These are some observations on the manuscript:

The ABSTRACT should provide more information on the "core regional virome".

The ABSTRACT should include the size of samples analyzed or the main genera of greatest abundance.

The ABSTRACT should highlight whether or not significant arboviruses were detected.

Taking into account the findings of the study, it is suggested that the wording and structure of the objective be improved.

The introduction and/or methodology should better justify why the departments were chosen. If there are previous reports of arboviruses other than Dengue, Zika or Chicungunya transmitted by vectors.

They should justify more details of the sampling effort and specify sampling days in each department, specifying specific dates.

What is the justification for working with such high sample pools (n= 50), taking into account the sensitivity and depth of the analysis method?

It is suggested to better detail the databases used for taxonomic assignments.

Explain and improve how the taxonomic confirmation of the host was performed. There are some highly complex groups of mosquitoes by morphology alone. Were metagenomic sequencing data also used?

In the discussion when citing viruses by species and contrasting them with those reported in the literature, a table can be generated that integrates more ecological variables associated with the species.

The size of the images is too small, it is recommended to improve it.

Separate into separate images or files "Heatmap and Non-metric Multidimensional Scaling (NMDS)" of the last figure.

### REVIEWER #2

Reviewer comments:

The manuscript "Viral metagenomics in mosquitoes as potential vectors of arboviruses in the Colombian Caribbean: characterization of a "core" regional virome" presents a comprehensive study characterizing the viral diversity of mosquitoes in the Colombian Caribbean using a metagenomic approach. It is quite original and might be an important contribution to the field. However, there are some issues that need to be addressed before the manuscript can be recommended for publication.

Introduction

The introduction is well written and covers the necessary information to present the topic. I would recommend the change of International studies to World wide studies in line 25 on page 5.

M and M

The authors should consider rewriting the first paragraph where the reasoning for choosing the sampling points. It is a little bit repetitive.

There are some typos on line 40, 50 and 52 on page 6

The authors present the RNA extraction protocol for isolating viral RNA. What about DNA viruses that might be present in the sampled mosquitoes? Their DNA can be extracted with the same protocol or the authors are not considering the abundance of such viruses? Please explain.

Why the authors have chosen to use Bray curtis to assess Beta diversity? There are a bunch of other tests to be performed to assess Beta diversity? Please explain

Results

The first paragraph should be rewritten … it is misleading and confusing ( I. e. on line 21 the sentence is disconnected by the inappropriate use of whereas).

The authors mention that "Mansonia titillans presented 38 viral species, of which 17 had no taxonomic classification, while 21 viruses belonged to the orders Picornavirales, Bunyavirales, and Mononegavirales, but could not be placed within the families of these orders (Figure 1)". There is no such information on Figure 1. Please provide a proper figure for that.

Colombia, September 2, 2025
Journal Memórias do Instituto Oswaldo Cruz
Ref. Corrections to the manuscript MIOC-2025-0131

Dear editor, thank you for considering our research for publication in your prestigious journal.

The manuscript ID MIOC-2025-0131 entitled "Viral metagenomics in mosquitoes as potential vectors of arboviruses in the Colombian Caribbean: characterization of a "core" regional virome," which we submitted to the Memórias do Instituto Oswaldo Cruz, has been fully reviewed by authors.

According to the suggestions of the reviewers, we responded point by point to the reviewer's comments. Furthermore, a new version of the manuscript, including tables and figure legends, has been enclosed.

Yours faithfully,
Richard Hoyos
Salim Mattar
Corresponding authors

Reviewer: 1
Reviewer comments. These are some observations on the manuscript:
1. The ABSTRACT should provide more information on the "core regional virome".

The ABSTRACT should include the size of samples analyzed or the main genera of greatest abundance. The ABSTRACT should highlight whether or not significant arboviruses were detected.

R/ The abstract has been modified according to the reviewers' comments, as shown below:

2nd Page - Lines 26 - 43:

This study characterized the RNA virome of mosquitoes in the Colombian Caribbean, focusing on the core regional virome and its role in the dynamics of arboviruses. Mosquitoes are critical vectors in tropical regions where arboviruses like Dengue and Zika are prevalent. In 2023, 4,074 mosquitoes from genera Mansonia, Coquillettidia, and Anopheles were collected across Córdoba, Sucre, Bolívar, and Magdalena during rainy and dry seasons. Specimens, pooled in groups of 50, underwent RNA extraction and sequencing on the MGI-G50™ platform. Bioinformatic analyses using the DIAMOND-MEGANizer pipeline and R packages (phyloseq, vegan, ggplot2) identified 22 viral families and 24 unclassified RNA viruses. The core regional virome, defined as viruses consistently present across species and seasons, was dominated by insect-specific viruses (ISVs) like Aedes aegypti to virus 1 and 2, Astopletus, and Cumbaru, alongside Picornaviridae (30% of reads), Rhabdoviridae (20%), Orthomyxoviridae, and Bunyavirales. Mansonia titillans (38 species) and Coquillettidia nigricans (21 species) showed the highest viral richness. No significant arboviruses were detected, highlighting ISV dominance. Virome composition varied seasonally, with greater diversity in the rainy season due to increased breeding site availability and temperature. The core viromes stability suggests it modulates vector competence, potentially reducing arbovirus transmission. These findings advocate the use of metagenomics for enhanced vector surveillance and biological control strategies in neotropical ecosystems.

2. Taking into account the findings of the study, it is suggested that the wording and structure of the objective be improved.

R/ Page 5 - Lines 118 - 121:

The objective was modified according to the reviewer's suggestions, as follows: This study aimed to characterize the virome of mosquito vectors of arboviruses, such as Mansonia titillans, Coquillettidia nigricans, Anopheles albimanus, Anopheles darlingi, Culex nigripalpus, and Culex quinquefasciatus, which are highly abundant in ecosystems related to the Colombian Caribbean, through metagenomic sequencing.

3. The introduction and/or methodology should better justify why the departments were chosen. If there are previous reports of arboviruses other than Dengue, Zika or Chicungunya transmitted by vectors.

R/ The introduction was modified following the suggestion of the reviewer. Page 4, lines 93 - 107:

The Colombian Caribbean is a strategic region for arbovirus research due to its high diversity of mosquito vectors —including Aedes, Anopheles, Culex, Mansonia, and Coquillettidia— and the heterogeneity of its habitats, such as wetlands, mangroves, gallery forests, agricultural areas, and peri-urban environments (3,6,8,10,12,14). These ecological conditions favor the persistence and transmission of emerging and re-emerging viruses, and facilitate interactions between mosquitoes and a wide range of potential reservoirs, including migratory birds —implicated in the spread of WNV and SLEV across the continent—, bats, rodents, and other wild mammals (1,3,5,8,10-14). In this sense, the departments of Cordoba, Sucre, Bolivar, and Magdalena, combine factors that make them a priority for entomovirological surveillance: (i) a history of circulation of arboviruses other than DENV, ZIKV, and CHIKV, including serological and molecular reports of WNV, SLEV, and YFV; (ii) the presence of aquatic ecosystems and flood-prone areas that sustain large mosquito populations; (iii) intense interaction between natural areas and agricultural or livestock landscapes, increasing human-vector contact; and (iv) their location within an ecological and migratory connectivity corridor that may facilitate the introduction and spread of pathogens (28, 29).

R/ The methodology was modified following the suggestions.

Page 5, lines 127 - 143:

These sites were selected based on a combination of ecological, epidemiological, and land use criteria. Ecologically, they encompass a variety of habitats, including wetlands, mangroves, gallery forests, agricultural lands, and peri-urban areas, which sustain high mosquito diversity (Aedes, Anopheles, Culex, Mansonia, and Coquillettidia) and provide breeding conditions favorable for arbovirus circulation. Epidemiologically, these departments have documented the presence of arboviruses beyond DENV, ZIKV, and CHIKV, including serological and molecular evidence of West Nile virus (WNV), Saint Louis encephalitis virus (SLEV), and yellow fever virus (YFV), reflecting their potential as hotspots for emerging and re-emerging vector-borne pathogens (3,5,10).

The selected sites are situated along important migratory bird routes and contain diverse vertebrate fauna, such as bats, rodents, and other mammals, which may act as reservoirs or amplifying hosts for arboviruses (11-13). Anthropogenic factors, including wetland fragmentation, rice and monoculture expansion, and cattle ranching, increase human–vector contact and may enhance the likelihood of viral spillover (3,5). Additionally, the land-use patterns, agricultural activity, and presence of aquatic ecosystems prone to seasonal flooding in the departments create ideal conditions for sustaining large mosquito populations (14).

4. They should justify more details of the sampling effort and specify sampling days in each department, specifying specific dates.

R/ This section was modified following the suggestion. Page 6, lines 143 – 153

Vector sampling was conducted between February and October 2023, with two main field campaigns in each department timed to coincide with the transitional periods before and after the rainy and dry seasons characteristic of the Caribbean climate. The dry-to-rainy season transition was sampled between February and April 2023, and the rainy-to-dry season transition was sampled between September and October 2023.

At each location, the sampling effort included eight CDC light traps operated for 12 h during the night (18:00–06:00), complemented by active searches carried out by teams of 3–4 trained personnel using mouth aspirators at potential resting sites among vegetation during early morning (06:00–10:00) and late afternoon (15:00–18:00) hours. Additionally, a Shannon trap was operated from 18:00 to 21:00, during which the same 3–4 trained personnel performed active mosquito collections using mouth aspirators to capture the specimens attracted to the light.

5. What is the justification for working with such high sample pools (n= 50), taking into account the sensitivity and depth of the analysis method?

R/ The decision to use pools of 50 mosquitoes per sample was based on fundamental considerations aligned with the objectives of our study.

Sensitivity for detecting low-abundance viruses. Viral loads in individual mosquitoes are often low, especially for viruses that are non-pathogenic to the insect or are in the early stages of infection. Viral RNA may be below the detection limit of metagenomic sequencing in individual samples. Pooling 50 mosquitoes significantly increased the probability of capturing sufficient viral genetic material. This is crucial for obtaining the most comprehensive view of the virome.

Obtaining sufficient high-quality genetic material. RNA metagenomic sequencing (especially for environmental/vector samples) requires a certain amount of total RNA input to build high-quality libraries and achieve a meaningful sequencing depth. Extracting RNA from a single mosquito does not yield the quantity or quality necessary to build robust metagenomic libraries, increasing the risk of technical bias and low coverage.

Logistical feasibility and population-level approaches. Our main objective was to characterize the virome present in mosquito populations from four municipalities in the Colombian Caribbean. Pool sampling is an efficient and widely accepted strategy for metagenomic surveillance studies at the population or geographic scale (such as ours), where the focus is on overall diversity rather than on individual prevalence. Sampling and processing mosquitoes individually to achieve the same statistical sensitivity would have been logistically unfeasible within the scope of this initial study, especially considering the four geographically separated sites and the intensive processing and extraction protocols involved.

We acknowledge that working with large pools may dilute very rare viruses present in only one or a few individuals, making precise prevalence estimation more difficult. However, we implemented rigorous methodological and bioinformatic controls. We sequenced at a sufficient depth to capture viral signals, even within a large amount of host RNA reads. In addition, we employed site- and species-specific replicates, which increased the likelihood of detecting less prevalent viruses. Finally, the study prioritized identifying the range of viruses present over determining the absolute viral load or exact prevalence, objectives that require different designs (e.g., individual sampling + qPCR).

6. It is suggested to better detail the databases used for taxonomic assignments.

R/ We used the NCBI nr (non-redundant protein sequences) database, which was last modified on February 7, 2024. This comprehensive database contains protein sequences from multiple sources, including GenBank, RefSeq, PDB, Swiss-Prot, PIR, and PRF databases. Its extensive coverage is crucial for the unbiased identification of viruses, including novel or poorly characterized viruses that are not yet curated in specialized viral databases but whose homologous sequences may be present in the GenBank. This is a standard practice in exploratory virome metagenomic studies.

7. Explain and improve how the taxonomic confirmation of the host was performed. There are some highly complex groups of mosquitoes by morphology alone. Were metagenomic sequencing data also used?

R/ Mosquitoes were identified to the lowest possible taxonomic level using specialized morphological keys for Neotropical Culicidae (30–39) under stereomicroscopes on chilled trays in a climate-controlled room (16 °C) to preserve the diagnostic structures. Identification was performed independently by experienced entomologists, and any discrepancies were resolved by consensus agreement. After this initial identification, representative specimens of each morphospecies were set aside and re-examined to verify key diagnostic characteristics, ensuring consistency and accuracy across samples. For morphologically challenging groups (e.g., Culex subgenus Melanoconion), examination included multiple diagnostic traits, and when male specimens were available, the genitalia were dissected and analyzed to confirm species-level identifications.

Although host confirmation was primarily based on morphology, metagenomic sequencing data were also screened for mosquito-specific genetic markers. A subset of quality-filtered reads from each pool was aligned against reference mitochondrial cytochrome oxidase I (COI) sequences and whole-genome scaffolds of relevant mosquito species available in the NCBI database. This cross-validation step was particularly useful for species complexes and ecologically overlapping taxa, and in all cases, morphological and molecular identifications were congruent, supporting the accuracy of the host assignments.

The paragraph on taxonomic identification of mosquitoes was modified as follows, Page 6, lines 162 - 176:

"Identification was carried out using specialized taxonomic keys for Neotropical Culicidae (30–39) in a climate-controlled room (16 °C) under stereomicroscopes, with specimens placed on chilled trays to preserve their morphological integrity. Mosquitoes were identified to the lowest possible taxonomic level, and representative specimens of each morphospecies were set aside and re-examined to verify key diagnostic characters, ensuring consistency and accuracy across all samples. For morphologically challenging groups (e.g., Culex subgenus Melanoconion), identification included the evaluation of multiple diagnostic traits, and when male specimens were available, the genitalia were dissected and analyzed to confirm species-level assignments. Although morphological identification served as the primary method for host confirmation, metagenomic sequencing data were screened for mosquito-specific genetic markers. A subset of quality-filtered reads from each pool was aligned (data not shown) against reference mitochondrial cytochrome oxidase I (COI) sequences and whole-genome scaffolds of relevant mosquito species available in the NCBI database. This cross-validation step was particularly useful for species complexes and ecologically overlapping taxa in this study.

8. In the discussion when citing viruses by species and contrasting them with those reported in the literature, a table can be generated that integrates more ecological variables associated with the species.

R/ The table was generated and used as a supplementary material (Table S1). Page 13, line 354.

9. The size of the images is too small, it is recommended to improve it.

R/ The suggestion was accepted.

10. Separate into separate images or files "Heatmap and Non-metric Multidimensional Scaling (NMDS)" of the last figure.

R/ The suggestion was accepted.

Reviewer: 2

Reviewer comments:

The manuscript "Viral metagenomics in mosquitoes as potential vectors of arboviruses in the Colombian Caribbean: characterization of a "core" regional virome" presents a comprehensive study characterizing the viral diversity of mosquitoes in the Colombian Caribbean using a metagenomic approach. It is quite original and might be an important contribution to the field. However, there are some issues that need to be addressed before the manuscript can be recommended for publication.

1. Introduction

The introduction is well written and covers the necessary information to present the topic. I would recommend the change of International studies to World wide studies in line 25 on page 5.

R/ Thank you for the correction, suggestion was accepted.

M and M

The authors should consider rewriting the first paragraph where the reasoning for choosing the sampling points. It is a little bit repetitive.

R/ The methodology was modified following the suggestion. Page 5, lines 127 -142:

These sites were selected based on a combination of ecological, epidemiological, and land-use criteria. Ecologically, they encompass a variety of habitats —including wetlands, mangroves, gallery forests, agricultural lands, and peri-urban areas— that sustain high mosquito diversity (Aedes, Anopheles, Culex, Mansonia, Coquillettidia) and provide breeding conditions favorable for arbovirus circulation. Epidemiologically, these departments have documented the presence of arboviruses beyond DENV, ZIKV, and CHIKV, including serological and molecular evidence of West Nile virus (WNV), Saint Louis encephalitis virus (SLEV), and yellow fever virus (YFV), reflecting their potential as hotspots for emerging and re-emerging vector-borne pathogens (3,5,10).

The selected sites are also situated along important migratory bird routes and contain diverse vertebrate fauna such as bats, rodents, and other mammals, which may act as reservoirs or amplifying hosts for arboviruses (11-13). Anthropogenic factors, including wetland fragmentation, rice and monoculture expansion, and cattle ranching, increase human–vector contact and may enhance the likelihood of viral spillover (3,5). Additionally, the departments' land-use patterns, agricultural activity, and presence of aquatic ecosystems prone to seasonal flooding create ideal conditions for sustaining large mosquito populations (14).

There are some typos on line 40, 50 and 52 on page 6.

R/ Thank you, the text was modified

The authors present the RNA extraction protocol for isolating viral RNA. What about DNA viruses that might be present in the sampled mosquitoes? Their DNA can be extracted with the same protocol or the authors are not considering the abundance of such viruses? Please explain.

R/ We thank the reviewer for pointing this out to us. The extraction protocol used was specifically optimized for total RNA, which has direct implications for the ability to detect DNA viruses. However, the primary objective of this study was to characterize the RNA virome of mosquitoes from the Colombian Caribbean. This decision was based on epidemiological considerations. Mosquitoes are the primary vectors of RNA viruses with a significant impact on human and animal health in the region (e.g., Dengue, Zika, Chikungunya, West Nile fever, Venezuelan equine encephalitis, and Orthobunyaviruses such as Oropouche). Our approach aimed to maximize the detection of relevant pathogens and other less-characterized RNA viruses.

The section "RNA Extraction, library preparation, and sequencing" was modified as  follows,page 7, lines 184 - 187:

RNA Extraction, library preparation, and sequencing. Mosquito pools of 50 individuals were used to ensure sufficient viral RNA detection and highquality metagenomic libraries. Pooling enhances sensitivity for low-abundance viruses, which may be below detection limits in individual mosquitoes, and provides adequate RNA yield for robust sequencing.

Why the authors have chosen to use Bray curtis to assess Beta diversity? There are a bunch of other tests to be performed to assess Beta diversity? Please explain

R/ We thank the reviewers for their relevant methodological observations. The Bray–Curtis index was chosen to assess beta diversity based on specific technical, biological, and practical reasons aligned with the nature of our data and the objectives of the study. Bray–Curtis is a metric that considers both the presence/absence and relative abundance of species. This is crucial in viromes, where some viruses may be highly abundant (e.g., endemic pathogens), whereas others may be rare but ecologically relevant. Ignoring abundance would underestimate the key ecological differences between samples. Similarly, virome data are often extremely sparse (i.e., low prevalence of most viruses). Bray–Curtis is less sensitive to zero values than other metrics, such as the Euclidean distance, which can overestimate differences in datasets with high sparsity. Why not use other metrics? UniFrac requires a robust phylogeny of sequences. Many sequences correspond to novel viruses with no close relatives, and taxonomic assignments are often at the family or genus level, which is insufficient for constructing accurate phylogenetic trees. UniFrac is ideal for bacteria (with conserved 16S rRNA) but impractical for exploratory viromes with a high proportion of unknown viruses. The Jaccard index considers only the presence or absence of species, ignoring differences in abundance. In contrast, Euclidean distance is highly sensitive to data sparsity, which can overestimate the differences between samples with low overlap of novel viruses.

Beta diversity was assessed using Bray-Curtis dissimilarity indices to measure differences in viral community composition between samples. The Bray-Curtis dissimilarity index was chosen for beta diversity to account for both the presence/absence and relative abundance of viruses, which is critical for viromes with high abundance variation and sparsity.

Results

The first paragraph should be rewritten … it is misleading and confusing ( I. e. on line 21 the sentence is disconnected by the inappropriate use of whereas).

R/ The first paragraph was modified as follows: Page 2 - Abstract:

A total of 4,074 mosquitoes were collected, and the pools were organized into 33 samples corresponding to the mosquito species previously selected based on their abundance (Table 1). A total of 1,729 individuals were captured during the dry season and 2,345 during the rainy season. The most abundant species identified across the four locations in the Colombian Caribbean were Mansonia titillans, Coquillettidia nigricans, and Anopheles albimanus, and other relevant species were Anopheles darlingi, Culex nigripalpus, and Culex quinquefasciatus.

The authors mention that "Mansonia titillans presented 38 viral species, of which 17 had no taxonomic classification, while 21 viruses belonged to the orders Picornavirales, Bunyavirales, and Mononegavirales, but could not be placed within the families of these orders (Figure 1)". There is no such information on Figure 1. Please provide a proper figure for that.

R/ The figure was provided and included in this section.

## SECOND REVIEW ROUND

REVIEWERS' COMMENTS

### REVIEWER #1

Reviewer comments: The manuscript was thoroughly revised and all the reviewers' suggestions were addressed. The article is original, they have modern tools for virome diagnosis, and the results are well presented. The approach and results achieved support the scope of the objectives. The methodology changed substantially.

### REVIEWER #2

Reviewer comments: after careful consideration and taking into account the author's reply for the first round of revision, in my point of view the manuscript is suitable for publication

