## [Reviewer Report · FIRST REVIEW ROUND - REVIEWERS COMMENTS]

## REVIEWER #1

Reviewer comments:

These are some observations on the manuscript:

The ABSTRACT should provide more information on the “core regional virome”.

The ABSTRACT should include the size of samples analyzed or the main genera of greatest abundance.

The ABSTRACT should highlight whether or not significant arboviruses were detected.

Taking into account the findings of the study, it is suggested that the wording and structure of the objective be improved.

The introduction and/or methodology should better justify why the departments were chosen. If there are previous reports of arboviruses other than Dengue, Zika or Chicungunya transmitted by vectors.

They should justify more details of the sampling effort and specify sampling days in each department, specifying specific dates.

What is the justification for working with such high sample pools (n= 50), taking into account the sensitivity and depth of the analysis method?

It is suggested to better detail the databases used for taxonomic assignments.

Explain and improve how the taxonomic confirmation of the host was performed. There are some highly complex groups of mosquitoes by morphology alone. Were metagenomic sequencing data also used?

In the discussion when citing viruses by species and contrasting them with those reported in the literature, a table can be generated that integrates more ecological variables associated with the species.

The size of the images is too small, it is recommended to improve it.

Separate into separate images or files “Heatmap and Non-metric Multidimensional Scaling (NMDS)” of the last figure.

## REVIEWER #2

Reviewer comments:

The manuscript “Viral metagenomics in mosquitoes as potential vectors of arboviruses in the Colombian Caribbean: characterization of a “core” regional virome” presents a comprehensive study characterizing the viral diversity of mosquitoes in the Colombian Caribbean using a metagenomic approach. It is quite original and might be an important contribution to the field. However, there are some issues that need to be addressed before the manuscript can be recommended for publication.

Introduction

The introduction is well written and covers the necessary information to present the topic. I would recommend the change of International studies to World wide studies in line 25 on page 5.

M and M

The authors should consider rewriting the first paragraph where the reasoning for choosing the sampling points. It is a little bit repetitive.

There are some typos on line 40, 50 and 52 on page 6

The authors present the RNA extraction protocol for isolating viral RNA. What about DNA viruses that might be present in the sampled mosquitoes? Their DNA can be extracted with the same protocol or the authors are not considering the abundance of such viruses? Please explain.

Why the authors have chosen to use Bray curtis to assess Beta diversity? There are a bunch of other tests to be performed to assess Beta diversity? Please explain

Results

The first paragraph should be rewritten … it is misleading and confusing ( I. e. on line 21 the sentence is disconnected by the inappropriate use of whereas).

The authors mention that “Mansonia titillans presented 38 viral species, of which 17 had no taxonomic classification, while 21 viruses belonged to the orders Picornavirales, Bunyavirales, and Mononegavirales, but could not be placed within the families of these orders (Figure 1)”. There is no such information on Figure 1. Please provide a proper figure for that.

---

## [Author Response · AUTHORS RESPONSE TO REVIEWERS]

## AUTHORS’ RESPONSE TO THE REVIEWERS

Colombia, September 2, 2025

Journal Memórias do Instituto Oswaldo Cruz

Ref. Corrections to the manuscript MIOC-2025-0131

Dear editor, thank you for considering our research for publication in your prestigious journal.

The manuscript ID MIOC-2025-0131 entitled “Viral metagenomics in mosquitoes as potential vectors of arboviruses in the Colombian Caribbean: characterization of a “core” regional virome,” which we submitted to the Memórias do Instituto Oswaldo Cruz, has been fully reviewed by authors.

According to the suggestions of the reviewers, we responded point by point to the reviewer’s comments. Furthermore, a new version of the manuscript, including tables and figure legends, has been enclosed.

Yours faithfully,

Richard Hoyos

Salim Mattar

Corresponding authors

## Reviewer: 1

Reviewer comments. These are some observations on the manuscript:

1. The ABSTRACT should provide more information on the “core regional virome”.

The ABSTRACT should include the size of samples analyzed or the main genera of greatest abundance. The ABSTRACT should highlight whether or not significant arboviruses were detected.

R/ The abstract has been modified according to the reviewers’ comments, as shown below:

2nd Page - Lines 26 - 43:

This study characterized the RNA virome of mosquitoes in the Colombian Caribbean, focusing on the core regional virome and its role in the dynamics of arboviruses. Mosquitoes are critical vectors in tropical regions where arboviruses like Dengue and Zika are prevalent. In 2023, 4,074 mosquitoes from genera Mansonia, Coquillettidia, and Anopheles were collected across Córdoba, Sucre, Bolívar, and Magdalena during rainy and dry seasons. Specimens, pooled in groups of 50, underwent RNA extraction and sequencing on the MGI-G50™ platform. Bioinformatic analyses using the DIAMOND-MEGANizer pipeline and R packages (phyloseq, vegan, ggplot2) identified 22 viral families and 24 unclassified RNA viruses. The core regional virome, defined as viruses consistently present across species and seasons, was dominated by insect-specific viruses (ISVs) like Aedes aegypti to virus 1 and 2, Astopletus, and Cumbaru, alongside Picornaviridae (30% of reads), Rhabdoviridae (20%), Orthomyxoviridae, and Bunyavirales. Mansonia titillans (38 species) and Coquillettidia nigricans (21 species) showed the highest viral richness. No significant arboviruses were detected, highlighting ISV dominance. Virome composition varied seasonally, with greater diversity in the rainy season due to increased breeding site availability and temperature. The core viromes stability suggests it modulates vector competence, potentially reducing arbovirus transmission. These findings advocate the use of metagenomics for enhanced vector surveillance and biological control strategies in neotropical ecosystems.

2. Taking into account the findings of the study, it is suggested that the wording and structure of the objective be improved.

R/ Page 5 - Lines 118 - 121:

The objective was modified according to the reviewer’s suggestions, as follows: This study aimed to characterize the virome of mosquito vectors of arboviruses, such as Mansonia titillans, Coquillettidia nigricans, Anopheles albimanus, Anopheles darlingi, Culex nigripalpus, and Culex quinquefasciatus, which are highly abundant in ecosystems related to the Colombian Caribbean, through metagenomic sequencing.

3. The introduction and/or methodology should better justify why the departments were chosen. If there are previous reports of arboviruses other than Dengue, Zika or Chicungunya transmitted by vectors.

R/ The introduction was modified following the suggestion of the reviewer. Page 4, lines 93 - 107:

The Colombian Caribbean is a strategic region for arbovirus research due to its high diversity of mosquito vectors —including Aedes, Anopheles, Culex, Mansonia, and Coquillettidia— and the heterogeneity of its habitats, such as wetlands, mangroves, gallery forests, agricultural areas, and peri-urban environments (3,6,8,10,12,14). These ecological conditions favor the persistence and transmission of emerging and re-emerging viruses, and facilitate interactions between mosquitoes and a wide range of potential reservoirs, including migratory birds —implicated in the spread of WNV and SLEV across the continent—, bats, rodents, and other wild mammals (1,3,5,8,10-14). In this sense, the departments of Cordoba, Sucre, Bolivar, and Magdalena, combine factors that make them a priority for entomovirological surveillance: (i) a history of circulation of arboviruses other than DENV, ZIKV, and CHIKV, including serological and molecular reports of WNV, SLEV, and YFV; (ii) the presence of aquatic ecosystems and flood-prone areas that sustain large mosquito populations; (iii) intense interaction between natural areas and agricultural or livestock landscapes, increasing human-vector contact; and (iv) their location within an ecological and migratory connectivity corridor that may facilitate the introduction and spread of pathogens (28, 29).

R/ The methodology was modified following the suggestions.

Page 5, lines 127 - 143:

These sites were selected based on a combination of ecological, epidemiological, and land use criteria. Ecologically, they encompass a variety of habitats, including wetlands, mangroves, gallery forests, agricultural lands, and peri-urban areas, which sustain high mosquito diversity (Aedes, Anopheles, Culex, Mansonia, and Coquillettidia) and provide breeding conditions favorable for arbovirus circulation. Epidemiologically, these departments have documented the presence of arboviruses beyond DENV, ZIKV, and CHIKV, including serological and molecular evidence of West Nile virus (WNV), Saint Louis encephalitis virus (SLEV), and yellow fever virus (YFV), reflecting their potential as hotspots for emerging and re-emerging vector-borne pathogens (3,5,10).

The selected sites are situated along important migratory bird routes and contain diverse vertebrate fauna, such as bats, rodents, and other mammals, which may act as reservoirs or amplifying hosts for arboviruses (11-13). Anthropogenic factors, including wetland fragmentation, rice and monoculture expansion, and cattle ranching, increase human–vector contact and may enhance the likelihood of viral spillover (3,5). Additionally, the land-use patterns, agricultural activity, and presence of aquatic ecosystems prone to seasonal flooding in the departments create ideal conditions for sustaining large mosquito populations (14).

4. They should justify more details of the sampling effort and specify sampling days in each department, specifying specific dates.

R/ This section was modified following the suggestion. Page 6, lines 143 – 153

Vector sampling was conducted between February and October 2023, with two main field campaigns in each department timed to coincide with the transitional periods before and after the rainy and dry seasons characteristic of the Caribbean climate. The dry-to-rainy season transition was sampled between February and April 2023, and the rainy-to-dry season transition was sampled between September and October 2023.

At each location, the sampling effort included eight CDC light traps operated for 12 h during the night (18:00–06:00), complemented by active searches carried out by teams of 3–4 trained personnel using mouth aspirators at potential resting sites among vegetation during early morning (06:00–10:00) and late afternoon (15:00–18:00) hours. Additionally, a Shannon trap was operated from 18:00 to 21:00, during which the same 3–4 trained personnel performed active mosquito collections using mouth aspirators to capture the specimens attracted to the light.

5. What is the justification for working with such high sample pools (n= 50), taking into account the sensitivity and depth of the analysis method?

R/ The decision to use pools of 50 mosquitoes per sample was based on fundamental considerations aligned with the objectives of our study.

Sensitivity for detecting low-abundance viruses. Viral loads in individual mosquitoes are often low, especially for viruses that are non-pathogenic to the insect or are in the early stages of infection. Viral RNA may be below the detection limit of metagenomic sequencing in individual samples. Pooling 50 mosquitoes significantly increased the probability of capturing sufficient viral genetic material. This is crucial for obtaining the most comprehensive view of the virome.

Obtaining sufficient high-quality genetic material. RNA metagenomic sequencing (especially for environmental/vector samples) requires a certain amount of total RNA input to build high-quality libraries and achieve a meaningful sequencing depth. Extracting RNA from a single mosquito does not yield the quantity or quality necessary to build robust metagenomic libraries, increasing the risk of technical bias and low coverage.

Logistical feasibility and population-level approaches. Our main objective was to characterize the virome present in mosquito populations from four municipalities in the Colombian Caribbean. Pool sampling is an efficient and widely accepted strategy for metagenomic surveillance studies at the population or geographic scale (such as ours), where the focus is on overall diversity rather than on individual prevalence. Sampling and processing mosquitoes individually to achieve the same statistical sensitivity would have been logistically unfeasible within the scope of this initial study, especially considering the four geographically separated sites and the intensive processing and extraction protocols involved.

We acknowledge that working with large pools may dilute very rare viruses present in only one or a few individuals, making precise prevalence estimation more difficult. However, we implemented rigorous methodological and bioinformatic controls. We sequenced at a sufficient depth to capture viral signals, even within a large amount of host RNA reads. In addition, we employed site- and species-specific replicates, which increased the likelihood of detecting less prevalent viruses. Finally, the study prioritized identifying the range of viruses present over determining the absolute viral load or exact prevalence, objectives that require different designs (e.g., individual sampling + qPCR).

6. It is suggested to better detail the databases used for taxonomic assignments.

R/ We used the NCBI nr (non-redundant protein sequences) database, which was last modified on February 7, 2024. This comprehensive database contains protein sequences from multiple sources, including GenBank, RefSeq, PDB, Swiss-Prot, PIR, and PRF databases. Its extensive coverage is crucial for the unbiased identification of viruses, including novel or poorly characterized viruses that are not yet curated in specialized viral databases but whose homologous sequences may be present in the GenBank. This is a standard practice in exploratory virome metagenomic studies.

7. Explain and improve how the taxonomic confirmation of the host was performed. There are some highly complex groups of mosquitoes by morphology alone. Were metagenomic sequencing data also used?

R/ Mosquitoes were identified to the lowest possible taxonomic level using specialized morphological keys for Neotropical Culicidae (30–39) under stereomicroscopes on chilled trays in a climate-controlled room (16 °C) to preserve the diagnostic structures. Identification was performed independently by experienced entomologists, and any discrepancies were resolved by consensus agreement. After this initial identification, representative specimens of each morphospecies were set aside and re-examined to verify key diagnostic characteristics, ensuring consistency and accuracy across samples. For morphologically challenging groups (e.g., Culex subgenus Melanoconion), examination included multiple diagnostic traits, and when male specimens were available, the genitalia were dissected and analyzed to confirm species-level identifications.

Although host confirmation was primarily based on morphology, metagenomic sequencing data were also screened for mosquito-specific genetic markers. A subset of quality-filtered reads from each pool was aligned against reference mitochondrial cytochrome oxidase I (COI) sequences and whole-genome scaffolds of relevant mosquito species available in the NCBI database. This cross-validation step was particularly useful for species complexes and ecologically overlapping taxa, and in all cases, morphological and molecular identifications were congruent, supporting the accuracy of the host assignments.

The paragraph on taxonomic identification of mosquitoes was modified as follows, Page 6, lines 162 - 176:

“Identification was carried out using specialized taxonomic keys for Neotropical Culicidae (30–39) in a climate-controlled room (16 °C) under stereomicroscopes, with specimens placed on chilled trays to preserve their morphological integrity. Mosquitoes were identified to the lowest possible taxonomic level, and representative specimens of each morphospecies were set aside and re-examined to verify key diagnostic characters, ensuring consistency and accuracy across all samples. For morphologically challenging groups (e.g., Culex subgenus Melanoconion), identification included the evaluation of multiple diagnostic traits, and when male specimens were available, the genitalia were dissected and analyzed to confirm species-level assignments. Although morphological identification served as the primary method for host confirmation, metagenomic sequencing data were screened for mosquito-specific genetic markers. A subset of quality-filtered reads from each pool was aligned (data not shown) against reference mitochondrial cytochrome oxidase I (COI) sequences and whole-genome scaffolds of relevant mosquito species available in the NCBI database. This cross-validation step was particularly useful for species complexes and ecologically overlapping taxa in this study.

8. In the discussion when citing viruses by species and contrasting them with those reported in the literature, a table can be generated that integrates more ecological variables associated with the species.

R/ The table was generated and used as a supplementary material (Table S1). Page 13, line 354.

9. The size of the images is too small, it is recommended to improve it.

R/ The suggestion was accepted.

10. Separate into separate images or files “Heatmap and Non-metric Multidimensional Scaling (NMDS)” of the last figure.

R/ The suggestion was accepted.

## Reviewer: 2

Reviewer comments:

The manuscript “Viral metagenomics in mosquitoes as potential vectors of arboviruses in the Colombian Caribbean: characterization of a “core” regional virome” presents a comprehensive study characterizing the viral diversity of mosquitoes in the Colombian Caribbean using a metagenomic approach. It is quite original and might be an important contribution to the field. However, there are some issues that need to be addressed before the manuscript can be recommended for publication.

1. Introduction

The introduction is well written and covers the necessary information to present the topic. I would recommend the change of International studies to World wide studies in line 25 on page 5.

R/ Thank you for the correction, suggestion was accepted.

M and M

The authors should consider rewriting the first paragraph where the reasoning for choosing the sampling points. It is a little bit repetitive.

R/ The methodology was modified following the suggestion. Page 5, lines 127 -142:

These sites were selected based on a combination of ecological, epidemiological, and land-use criteria. Ecologically, they encompass a variety of habitats —including wetlands, mangroves, gallery forests, agricultural lands, and peri-urban areas— that sustain high mosquito diversity (Aedes, Anopheles, Culex, Mansonia, Coquillettidia) and provide breeding conditions favorable for arbovirus circulation. Epidemiologically, these departments have documented the presence of arboviruses beyond DENV, ZIKV, and CHIKV, including serological and molecular evidence of West Nile virus (WNV), Saint Louis encephalitis virus (SLEV), and yellow fever virus (YFV), reflecting their potential as hotspots for emerging and re-emerging vector-borne pathogens (3,5,10).

The selected sites are also situated along important migratory bird routes and contain diverse vertebrate fauna such as bats, rodents, and other mammals, which may act as reservoirs or amplifying hosts for arboviruses (11-13). Anthropogenic factors, including wetland fragmentation, rice and monoculture expansion, and cattle ranching, increase human–vector contact and may enhance the likelihood of viral spillover (3,5). Additionally, the departments’ land-use patterns, agricultural activity, and presence of aquatic ecosystems prone to seasonal flooding create ideal conditions for sustaining large mosquito populations (14).

There are some typos on line 40, 50 and 52 on page 6.

R/ Thank you, the text was modified

The authors present the RNA extraction protocol for isolating viral RNA. What about DNA viruses that might be present in the sampled mosquitoes? Their DNA can be extracted with the same protocol or the authors are not considering the abundance of such viruses? Please explain.

R/ We thank the reviewer for pointing this out to us. The extraction protocol used was specifically optimized for total RNA, which has direct implications for the ability to detect DNA viruses. However, the primary objective of this study was to characterize the RNA virome of mosquitoes from the Colombian Caribbean. This decision was based on epidemiological considerations. Mosquitoes are the primary vectors of RNA viruses with a significant impact on human and animal health in the region (e.g., Dengue, Zika, Chikungunya, West Nile fever, Venezuelan equine encephalitis, and Orthobunyaviruses such as Oropouche). Our approach aimed to maximize the detection of relevant pathogens and other less-characterized RNA viruses.

The section “RNA Extraction, library preparation, and sequencing” was modified as follows,page 7, lines 184 - 187:

RNA Extraction, library preparation, and sequencing. Mosquito pools of 50 individuals were used to ensure sufficient viral RNA detection and highquality metagenomic libraries. Pooling enhances sensitivity for low-abundance viruses, which may be below detection limits in individual mosquitoes, and provides adequate RNA yield for robust sequencing.

Why the authors have chosen to use Bray curtis to assess Beta diversity? There are a bunch of other tests to be performed to assess Beta diversity? Please explain

R/ We thank the reviewers for their relevant methodological observations. The Bray–Curtis index was chosen to assess beta diversity based on specific technical, biological, and practical reasons aligned with the nature of our data and the objectives of the study. Bray–Curtis is a metric that considers both the presence/absence and relative abundance of species. This is crucial in viromes, where some viruses may be highly abundant (e.g., endemic pathogens), whereas others may be rare but ecologically relevant. Ignoring abundance would underestimate the key ecological differences between samples. Similarly, virome data are often extremely sparse (i.e., low prevalence of most viruses). Bray–Curtis is less sensitive to zero values than other metrics, such as the Euclidean distance, which can overestimate differences in datasets with high sparsity. Why not use other metrics? UniFrac requires a robust phylogeny of sequences. Many sequences correspond to novel viruses with no close relatives, and taxonomic assignments are often at the family or genus level, which is insufficient for constructing accurate phylogenetic trees. UniFrac is ideal for bacteria (with conserved 16S rRNA) but impractical for exploratory viromes with a high proportion of unknown viruses. The Jaccard index considers only the presence or absence of species, ignoring differences in abundance. In contrast, Euclidean distance is highly sensitive to data sparsity, which can overestimate the differences between samples with low overlap of novel viruses.

Beta diversity was assessed using Bray-Curtis dissimilarity indices to measure differences in viral community composition between samples. The Bray-Curtis dissimilarity index was chosen for beta diversity to account for both the presence/absence and relative abundance of viruses, which is critical for viromes with high abundance variation and sparsity.

Results

The first paragraph should be rewritten … it is misleading and confusing ( I. e. on line 21 the sentence is disconnected by the inappropriate use of whereas).

R/ The first paragraph was modified as follows: Page 2 - Abstract:

A total of 4,074 mosquitoes were collected, and the pools were organized into 33 samples corresponding to the mosquito species previously selected based on their abundance (Table 1). A total of 1,729 individuals were captured during the dry season and 2,345 during the rainy season. The most abundant species identified across the four locations in the Colombian Caribbean were Mansonia titillans, Coquillettidia nigricans, and Anopheles albimanus, and other relevant species were Anopheles darlingi, Culex nigripalpus, and Culex quinquefasciatus.

The authors mention that “Mansonia titillans presented 38 viral species, of which 17 had no taxonomic classification, while 21 viruses belonged to the orders Picornavirales, Bunyavirales, and Mononegavirales, but could not be placed within the families of these orders (Figure 1)”. There is no such information on Figure 1. Please provide a proper figure for that.

R/ The figure was provided and included in this section.

---

## [Reviewer Report · REVIEWERS COMMENTS]

## REVIEWER #1

Reviewer comments: The manuscript was thoroughly revised and all the reviewers’ suggestions were addressed. The article is original, they have modern tools for virome diagnosis, and the results are well presented. The approach and results achieved support the scope of the objectives. The methodology changed substantially.

## REVIEWER #2

Reviewer comments: after careful consideration and taking into account the author´s reply for the first round of revision, in my point of view the manuscript is suitable for publication